# Human Papillomavirus Detected in Oropharyngeal Cancers from Chilean Subjects

**DOI:** 10.3390/v14061212

**Published:** 2022-06-02

**Authors:** Carolina Oliva, Diego Carrillo-Beltrán, Paul Boettiger, Iván Gallegos, Francisco Aguayo

**Affiliations:** 1Laboratorio de Oncovirología, Programa de Virología, Instituto de Ciencias Biomédicas (ICBM), Facultad de Medicina, Universidad de Chile, Santiago 8380000, Chile; diego.carrillo@uach.cl; 2Departamento de Otorrinolaringología, Hospital Clínico Universidad de Chile, Santiago 8380000, Chile; paulboettiger@gmail.com; 3Instituto de Bioquímica y Microbiología, Facultad de Ciencias, Universidad Austral de Chile, Valdivia 5090000, Chile; 4Departamento Anatomía Patológica, Hospital Clínico Universidad de Chile, Santiago 8380000, Chile; igallegosmendez@gmail.com; 5Advanced Center for Chronic Diseases (ACCDiS), Facultad de Medicina, Universidad de Chile, Santiago 8380000, Chile

**Keywords:** oropharyngeal cancer, human papillomavirus, head and neck, squamous cell carcinomas (OPSCC)

## Abstract

High-risk human papillomaviruses (HR-HPV) are the causal agents of an important subset of oropharyngeal cancers that has increased considerably in incidence in recent years. In this study, we evaluated the presence of HPV in 49 oropharyngeal cancers from Chilean subjects. The presence of HPV DNA was analyzed by conventional PCR, the genotypes were identified through sequencing, and the expression of E6/E7 transcripts was evaluated by a reverse transcriptase polymerase chain reaction (RT-PCR). Additionally, to determine p16 expression—a surrogate marker for oncogenic HPV infection—a tissue array was constructed for immunohistochemistry (IHC). HPV was detected in 61.2% of oropharyngeal carcinomas, the most prevalent genotype being HPV16 (80%). E6 and E7 transcripts were detected in 91.6% and 79.1% of the HPV16-positive specimens, respectively, demonstrating functional HPV infections. Furthermore, p16 expression was positive in 58.3% of cases. These findings show a high prevalence of HR-HPV in oropharyngeal tumors from Chile, suggesting the necessity of additional studies to address this growing public health concern.

## 1. Introduction

Human papillomavirus (HPV) is the main causal agent of the rise in oropharyngeal squamous cell carcinomas (OPSCCs) in developed countries, particularly in Northern Europe and the United States [1], with an approximate HPV prevalence of 25–70% [2,3]. It characteristically affects a population group of young men with less tobacco consumption than is typical for classic HPV (−) OPSCCs [4].

HPV is a member of the Papillomaviridae family of viruses, with icosahedral symmetry and a genome consisting of double-stranded closed circular DNA containing two coding regions for early (E) and late (L) products. The E region encodes for E1, E2, E4, E5, E6, and E7 proteins. E1 and E2 play a role in regulating viral DNA replication and gene expression; E6 and E7 are involved in increased cell proliferation and the transforming properties of HPV, while E4 and E5 are also involved in viral replication. The L region encodes for the major (L1) and minor (L2) viral capsid proteins [5]. Between the end of L1 and the start of the *E6* gene, the long control region (LCR) is located, a non-coding genomic segment that controls HPV gene expression and replication [6]. More than 200 HPV genotypes have been identified and characterized, functionally classified into types of high, intermediate, or low risk of cancer [7]. The HPV16 genotype belongs to the high risk (HR) subtype and can be found in over 70% of HR-HPV OPSCCs [8,9]. The virus infects the cells of the basal epithelium through localized wounds or micro-abrasions and its replication cycle is linked to the differentiation of the epithelial tissue of the host [10]. In the oropharynx, HPV-associated tumors occur mainly at the base of the tongue and palatine tonsils since the virus has a tropism for epithelial reticular cells present in the palatine crypts, which are susceptible to transformation and could be analogous to the cervical-uterine transformation zone [11]. The presence of HPV DNA has been determined in other head and neck carcinomas, such as the oral cavity [12,13], larynx [14,15] and paranasal cavities [16,17]; however, an etiological role of HPV in these tumors is not clear yet. Interestingly, HPV presence has been associated with a better survival prognosis, especially in rhinosinusal carcinomas [16,18].

One hallmark of HPV-driven tumors is the continuous expression of the E6 and E7 oncoproteins, frequently caused by integration of the viral episome into the host genome. Inhibition of these oncoproteins leads to cell cycle arrest and apoptosis, demonstrating the importance of E6 and E7 in maintaining the transformed phenotype [19]. The E6 oncoprotein recruits the E6-associated protein (E6AP), a ubiquitin ligase that triggers p53 ubiquitination and degradation, deregulating the G1/S and G2/M checkpoints, typically induced by DNA damage [5,19]. E6 also induces the expression and activity of hTERT, the catalytic subunit of telomerase, favoring cell immortalization. Meanwhile, E7 promotes cell transformation through degradation of the retinoblastoma protein (Rb), releasing the transcriptional factors E2F and subsequently stimulating the S phase, which promotes the overexpression of p16 protein, widely used as the surrogate marker for transcriptionally active HR-HPV infection in OPSCCs [20].

To date, there are no published studies on HR-HPV in OPSCCs in Chile. Thus, this study represents the first documentation of the presence and characterization of HPV in patients from Chile with oropharyngeal tumors.

## 2. Materials and Methods

### 2.1. Sample Collection

A retrospective study was carried out using biopsies fixed in formalin and embedded in paraffin (FFPE) with the diagnosis of OPSCC, stored in the Pathological Anatomy Service of the Hospital Clínico Universidad de Chile, from 2010 to 2020. The clinical data were obtained from the pathology reports and the patients’ clinical records. This study was approved by the Ethics Committee Board of the Clinical Hospital of the University of Chile (No. 47/19; 19 November 2019). Of the 51 samples initially collected, 49 were considered to have amplifiable DNA (see Results Section 3). Additionally, 31/49 were men and 18/49 were women with a mean age of 62.6 years. Of the patients, 20 were currently or former smokers, 12 patients were never smokers, and the smoking status was unknown for 17 patients. Regarding the pathological analysis, 33/49 samples had keratinized morphology, 24 samples were poorly differentiated, 16 samples were moderately differentiated, and 6 samples were well differentiated.

### 2.2. DNA Extraction, HPV Detection and Genotyping

To assess the presence of HPV DNA, FFPE tissue sections were incubated with digestion buffer (10 mM Tris-HCl pH 7.4, 0.5 mg/mL proteinase K, and 0.4% Tween 20) for 8 h at 56 °C with stirring. Subsequently, the samples were incubated at 95 °C for 10 min, centrifuged for 2 min at 14,000 rpm and stored at 4 °C until use. The amplification was performed by conventional PCR. First, the presence of a fragment of the *β-globin* gene was analyzed to determine the quality of the DNA with the following primers: PCO3 5′-ACACAACTGTGTTCACTAGC-3′ and PCO4 5′-CAACTTCATCCACGTTCACC-3′. The amplification program was as follows: denaturation at 95 °C for 5 min; 45 cycles with a cycling profile of 95 °C for 30 s, 52 °C for 30 s, and 72 °C for 30 s; and a final extension at 72 °C for 5 min. To identify the presence of HPV, we used the GP5+/GP6+ degenerate consensus primers for amplification of *L1* gene sequences [21]. We used conventional PCR with the following primers: GP5+ 5′-TTTGTTACTGTGGTAGATATCAC-3′ and GP6+ 5′-GAAAAATAAACTGTAAATCATATTC-3′. The PCR conditions were as follows: denaturation at 95 °C for 5 min; 45 cycles that included 1 min at 94 °C, 2 min at 50 °C, and 1.5 min at 72 °C; and a final elongation at 72 °C for 5 min. PCR products were characterized by 2.5% agarose gel electrophoresis, stained with SafeView PlusTM (abm, Vancouver, BC, Canada) and visualized under UV transillumination. To identify the HPV genotypes [22,23], GP5+/GP6+ PCR products were sent to Macrogen (Macrogen Co. Ltd., Seoul, Korea) for capillary electrophoresis sequencing (CES). Afterwards, the sequences obtained were analyzed using the NCBI Blast platform to determine the HPV genotype.

### 2.3. Tissue Microarray (TMA) Construction and p16 Immunohistochemistry

The TMAs were fabricated, including 5 mm oropharyngeal samples. The tissue microarrays were introduced into Benchmark Ultra equipment (Ventana, Medical systems Inc., Tucson, AZ, USA), and Tris-Borate-EDTA (TBE) buffer pH 8.0 was added for 92 min for antigen recovery. The microarrays were incubated at 37 °C in a humid chamber with the Cintec p16 antibody (Ventana, Medical systems Inc., Tucson, AZ, USA). The revealing of the antibody was carried out with the UltraView Dab Kit (Ventana, Medical systems Inc., Tucson, AZ, USA). The results for p16 IHC were interpreted by an experienced pathologist and were expressed as negative and positive according to the guidelines of the College of American Pathologists (positive with ≥70% nuclear and cytoplasmic staining) [24].

### 2.4. RNA Purification and Transcriptase-Reverse PCR

For the purification of RNA from FFPE, a High Pure RNA Paraffin Kit (Roche, Pleasanton, CA, USA) was used following the manufacturer’s instructions. Briefly, the RNA concentrations and purity were determined using a Nanodrop1000 spectrophotometer (NanoDrop Technologies, Wilmington, NC, USA). The cDNA was prepared in a 20 μL of reaction volume containing DNAse-treated RNA, 1 U of RNase inhibitor (Promega), 0.04 mg/mL of random primers (Promega, Madison, WI, USA), 2 mM of dNTP (Promega, Madison, WI, USA), and 10 U of Moloney murine leukemia virus (MMLV) reverse transcriptase (Promega, Madison, WI, USA). Levels of HPV16 E6 and E7 transcripts were evaluated by semi-quantitative RT-PCR using the following primers previously reported by us [25]: E6 small 16 F 5′–CTGCAAGCAACAGTTACTGCGA-3′ and E6 small 16 R 5′-TCACACACTGCATATGGATTCCC-3′, E7 small 16 F 5′-CAATATTGTAATGGGCTCTGTCC-3′ and E7 small 16 R 5′-ATTTGCAACCAGAGACAACTGAT-3′. RT-PCR for β-actin mRNA levels was used for the normalization of RNA expression using the following primers: F 5′-CCACACAGGGGAGGTGACTAG-3′ and R 5′-GGGCACGAAGGCTCATT-3. The amplification conditions were 94 °C for 5 min followed by 33 cycles of denaturation at 95 °C for 45 s, annealing at 56 °C for 40 s and extension at 72 °C for 45 s, with a final extension for 5 min at 72 °C. RT-PCR products were characterized by 2,5% agarose gel electrophoresis, stained with SafeView PlusTM (abm, Vancouver, BC, Canada) and visualized under UV transillumination (Vilber-Lourmat, Marne La Vallée, France). For semi-quantitative analysis, ImageJ software version 1.52a (National Institutes of Health, Bethesda, MD, USA) was used.

### 2.5. Statistical Analysis

Statistical analysis was performed using the GraphPad Prism 9 (Version 9.3.1) and Stata 17.0 (StataCorp, Texas, TX, USA) software. Results were considered statistically significant when *p* was less than or equal to 0.05.

## 3. Results

### 3.1. Clinical Pathological Characteristics

A total of 51 samples from 51 patients were collected, 2 of which were eliminated for not presenting detection of the β -Globin fragment by PCR. Of the 49 selected samples, 18/49 (37.7%) were female and 31/49 (63.3%) were male; the average age was 62.6 years with a range of 39–85. Additionally, 46.87% of the patients were currently smokers at the time of diagnosis. A total of 26 (53%) oropharyngeal specimens were obtained from the palatine tonsils, 14 (28.6%) were from the base of the tongue, and 9 cases (18.4%) were from the soft palate. Table 1 describes clinicopathological features according to anatomical site of OPSCCs. It was observed that carcinomas from the palatine tonsils tended to have poor-to-moderate differentiation compared to those from the tongue and soft palate (*p* = 0.0323). There were no differences regarding gender (*p* = 0.85), age (*p* = 0.3035), or tumor keratinization (*p* = 0.2115). Oropharyngeal carcinomas from the palatine tonsils presented higher p16 positivity compared to those from the tongue or soft palate (*p* < 0.0001). Figure 1 describes representative images of tumor sections used in the tissue arrays with hematoxylin–eosin staining and p16 IHC.

### 3.2. HPV Detection and Genotyping

It was observed that 30/49 samples were positive for HPV (61.2%) and 19/49 were negative (38.8%). PCR products from HPV (+) oropharyngeal samples were submitted to Macrogen for sequencing. The obtained sequences were processed using the NCBI Blast platform to determine the HPV genotype. Through this analysis, it was determined that 24 cases were HPV16 (80%), 2 cases were HPV33 (6.7%), 1 case was HPV18 (3.3%), and 3 cases were HPV6 (10%) (Table 2). Note that this method of analysis does not allow for the identification of co-infections with two or more HPV genotypes in the same specimen.

### 3.3. p16 Immunohistochemistry

It was detected that 58.3% (28/48) of the cases were p16-positive. Oropharyngeal carcinomas from the palatine tonsils presented higher p16 positivity compared to those from the tongue or soft palate with *p* < 0.0001 (Table 1). Figure 1 shows representative images of tumor sections used in the tissue arrays with p16-positive (C, F) and p16-negative (I) IHC. As previously mentioned, the palatine tonsils were highly positive for p16 (*p* < 0.0001). Additionally, we evaluated the correlation between p16 expression and HPV presence. As expected, there was a positive correlation between positive p16 and the presence of HPV DNA (*p* = 0.0004, Table 3). However, six p16-negative cases were HPV-positive, including three HPV6 and three HPV16 cases. Conversely, five p16-positive cases were HPV-negative.

### 3.4. E6/E7 Transcripts in OPSCCs

The presence of E6/E7 transcripts was analyzed by RT-PCR in HPV16-positive samples. The results showed that 22/24 (91.6%) expressed E6 transcripts and 19/24 (79.1%) specimens expressed detectable levels of E7 transcripts. All cases presenting E7 transcripts were also positive for E6 transcripts. Figure 2 shows representative examples of the expression of E6/E7 transcripts in a 2.5% agarose gel.

Agarose gel electrophoresis (2.5%) of products obtained after RT-PCR for HPV16 E6 and E7 gene expression. Left to right: 100 bp DNA ladder, palatine tonsil samples positive for HPV16, MMLV control (HPV16 E6-positive sample without MMLV treatment), negative control (nuclease-free water), and positive control (CasKi cells). Expected molecular weights are 96 bp and 110 bp for E6 and E7 transcripts, respectively.

## 4. Discussion

HR-HPVs are the causal agents of an important subset of OPSCCs whose prevalence has increased considerably in recent years, affecting a younger non-smoking population when compared to previous decades [1]. Furthermore, HPV (+) cases are different clinical–molecular entities when compared to HPV (−) OPSCCs. HPV presence is also the most important positive predictor of overall survival and response to treatment in patients with OPSCC [26,27].

The worldwide HPV prevalence in oropharyngeal carcinoma is approximately 25–30%, with a higher frequency found in developed countries [8]. In this study, HPV was detected in 61.2% of cases. In South America, there is a HPV prevalence of 17.9% in oropharyngeal tumors (95% CI 7.6–31.4); however, there are few studies on OPSCCs [28]. A recent study in Brazil reported that HPV was present in 59.1% of cases by PCR/p16 [29]. Although, previous studies in the same country showed a wide range of HPV detection (4.1 to 55%) with the use of different methods [30,31,32]. Additionally, most studies of HPV in head and neck cancers in Latin America focused on the oral cavity. Indeed, HR-HPV was previously detected in 11% of oral squamous cell carcinomas (OSCCs) from Chilean patients [33]. Therefore, considering these epidemiological data, our findings suggest that Chile has a high prevalence of HPV-associated oropharyngeal carcinomas. One factor that may explain the difference in the frequency of HPV observation between Brazil and Chile could be socioeconomic status, which has been addressed previously [34]. Brazilian studies with low HPV prevalence were carried out in public institutions; conversely, this study used biopsies from a private institution. Interestingly, De Cicco et al. [29] collected specimens from a private institution and reported an HPV frequency similar to this study. Another factor potentially involved may be related to the time of samples storage in the reports with low HPV prevalence, not only due to the possible degradation of the genetic material, but also because the samples were taken prior to the epidemiological increase in HPV (+) OPSCCs that occurred over the last two decades [35]. Another difficulty in comparing reports is the use of different methods for HPV detection which show different sensitivities and specificities [30,31].

Pertaining to the anatomical location of OPSCCs, HPV DNA has previously been detected in 47% of tonsil tumors and 18.5% of base-of-tongue tumors [1]. In this study, we identified HPV presence in 73% (19/26) of the palatine tonsil tumors, followed by 50% (7/14) in the case of base-of-tongue tumors, although there were not statistically significant differences. Additionally, most of the HPV (+) specimens showed poor-to-moderate differentiation, though HPV (+) OPSCCs are characterized by an immature appearance, with a high nucleus: cytoplasm ratio and a high number of mitoses, with a basaloid and “immature” morphology. Thus, it has been suggested that it is not necessary to assign a grade of tumor differentiation to HPV (+) OPSCCs, also considering their well-known good prognoses regardless of cytological appearance [20,24]. On the other hand, HPV (+) OPSCCs are characterized by presenting a non-keratinizing morphology similar to the normal epithelium of the palatine tonsils [36]. In fact, although the HPV (+) specimens analyzed in this study presented a higher percentage of non-keratinizing morphology, this result was not statistically significant. A possible limitation in the histopathological analysis is the small size of the soft palate biopsies, which may lose representativeness of the sample.

As previously mentioned, OPSCCs attributable to HPV have been characterized by affecting a younger epidemiological group than their HPV (−) counterparts [4,37,38]. In this report, 63.3% (19/30) of HPV (+) carcinomas were present in patients under 65 years of age, while only 50% (9/18) of HPV (−) carcinomas were from that same age group, although this difference was not statistically significant.

HPV16 was the most frequent HR-HPV genotype in the analyzed OPSCCs, accounting for 80% of the cases, which is consistent with the literature, which reports a prevalence of over 70% [8,9]. In this study, HPV33, another HR-HPV genotype [39,40], was detected in two cases (6.7%). Interestingly, in the study by Melo et al. [41], HPV33 was the most frequent genotype in low- and high-grade cervical lesions. We can observe this trend not only in the Chilean population, but also in a report from the Unites States, in which HPV33 was also the second most prevalent HPV genotype [9]. The difference in the prevalence of HPV genotypes in a population may be determined by the genotypic variants, which may differ in their oncogenic potential [42]. Among them, the HPV16 Asian American variant is one of the most aggressive [43]. It is important to have knowledge of the genotypes of each population to establish public health policies according to the local characteristics.

In this study, p16 positivity was 58.3% in OPSCCs from Chilean subjects. Indeed, p16 has been proposed as a surrogate biomarker which correlates with oncogenic HPV presence in OPSCCs [44]. For this reason, the 8^th^ edition of the staging system of the American Joint Committee on Cancer separates p16-positive and -negative OPSCCs into two different entities [26,45]. Additionally, there were six p16-negative cases which were HPV-positive, including three HPV6 cases, suggesting that this HPV genotype is a bystander. Additionally, three other cases were HPV16, suggesting the possibility that these cases were not HPV-driven cancers or that p16 was silenced by mutation or promoter hypermethylation in these patients, as previously reported in head and neck cancers [46,47]. Of note, five HPV-negative cases were p16-positive, suggesting the possibility of a false negative by PCR or that p16 was overexpressed by other mechanisms which are independent of E7 expression [48,49]. When analyzing E6/E7 expression in HPV16-positive OPSCCs, 19/24 (79.1%) of them expressed both transcripts. In addition, three samples expressed only the E6 transcript. Interestingly, of the five samples in which E6 and E7 transcripts were not detected, four of them were p16-negative, suggesting that even though the viral genome was present in the tissue, it was not playing an oncogenic role in the tumor. Both the detection of E6/E7 transcripts and the positivity for p16 by immunohistochemistry (overexpressed due to the degradation of the retinoblastoma protein by E7) suggest active oncogenic viral transcription and an oncogenic role in OPSCCs from Chile [50,51]. Although we used primers for the amplification of short regions of E6 and E7 mRNA, as suggested in FFPE tissues, it cannot be ruled out that the RNA has been partially fragmented or interrupted by both formalin fixation and storage time [20].

In conclusion, in this study we found a high HPV prevalence in OPSCCs from Chile, with significant correlation with p16 expression. More studies are warranted to dissect the public health consequences of these findings and establish control and treatment strategies for HPV-associated OPSCCs.

## Figures and Tables

**Figure 1 viruses-14-01212-f001:**
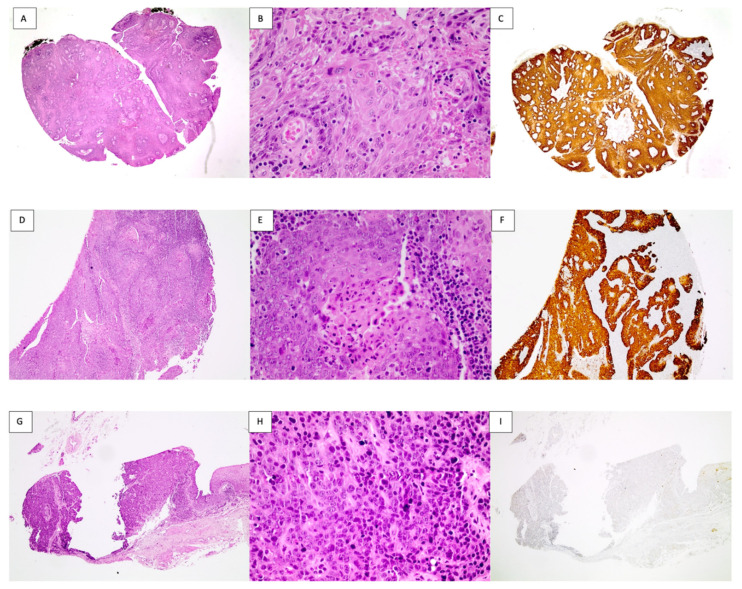
Representative images of the histological sections with hematoxylin–eosin staining and p16 IHC, of base of the tongue, palatine tonsil, and soft palate. (**A**–**C**): Tongue base squamous cell carcinoma. (**A**). Section with hematoxylin–eosin stain, tissue fragment showing infiltrating squamous cell carcinoma, partially keratinizing. (**B**). Detail of the cytological atypia with pleomorphism, hyperchromasia, and focal keratinization. (**C**). Intense and diffuse cytoplasmic and nuclear positivity for p16 in the neoplastic cells. (**D**–**F**): Palatine tonsils squamous cell carcinoma. (**D**). Section with hematoxylin–eosin stain, tonsillar tissue with loss of its architecture, and replacement by infiltrating squamous cell carcinoma. (**E**). Tumor nest showing less differentiated population towards the periphery, in the center with keratinization and partial necrosis accompanied by polymorphonuclear cells, lymphoplasmacytic inflammatory infiltrate in the vicinity. (**F**). Intense and diffuse cytoplasmic and nuclear positivity for p16 in the neoplastic cells. (**G**–**I**): Soft palate squamous cell carcinoma. (**G**). Section with hematoxylin–eosin stain, mucosa of the palate infiltrated by nonkeratinizing squamous cell carcinoma, with a small remnant of normal tissue towards the far right of the image. (**H**). Detail of the cytology with basaloid cells, hyperchromasia, and mitosis, without evidence of keratinization. (**I**). p16-negative specimen by IHC.

**Figure 2 viruses-14-01212-f002:**
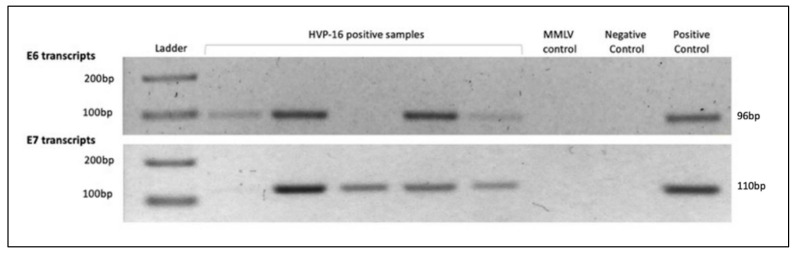
E6 and E7 transcripts in OPSCCs.

**Table 1 viruses-14-01212-t001:** Clinicopathological features of OPSCCs.

	Anatomical Site				
Features	Palatine Tonsils N (%)	Tongue Base N (%)	Soft Palate N (%)	Total	*p*-Value
**Gender**					*p* = 0.85
Female	9 (50)	6 (33.33)	3 (16.66)	18	
Male	17 (54.38)	8 (25.8)	6 (19.35)	31	
**Age**					*p* = 0.3035
≤65 years	18 (62.07)	7 (24.24)	4 (13.79)	29	
>65 years	8 (40)	7 (35)	5 (25)	20	
**Keratinization**					*p =* 0.2115
Keratinized	15 (45.45)	10 (30.3)	8 (24.24)	33	
Non-Keratinized	11 (68.75)	4 (25)	1 (6.25)	16	
**Differentation**					*p* = 0.0323 *
Poor	17 (69.56)	5 (21.73)	2 (8.26)	23	
Moderate	8 (47%)	4 (23.52)	5 (29.41)	17	
Well	1 (12.5)	5 (62.5)	2 (25)	8	
**p16 IHC**					*p* < 0.0001 ***
*Positive*	22 (78.57)	5 (17.85)	1 (3.57)	28	
*Negative*	3 (15)	9 (45)	8 (40)	20	

Gender, age, keratinization, differentiation, and p16 according to anatomical site. *: *p* ≤ 0.05: ***: *p* ≤ 0.001. *p*-value < 0.5 was considered statistically significant.

**Table 2 viruses-14-01212-t002:** HPV presence and genotypes in OPSCCs.

	Variable	Number of Cases (%)
*HPV Presence*	HPV-Negative	19 (38.8)
	HPV-Positive	30 (61.2)
*Oncogenic Risk*	Low	3 (10)
	High	27 (90)
*Genotype*	HPV6	3 (10)
	HPV16	24 (80)
	HVP18	1 (3.3)
	HVP33	2 (6.7)

HPV positivity was significantly associated with poor-to-moderate differentiation compared to HPV (−) samples (*p* = 0.029, Table 3). There was no statistically significant association between the presence of HPV and gender (*p* = 0.24), age (*p* = 0.3759), smoking status (*p* = 0.95), or anatomical location of the tumor (*p* = 0.19).

**Table 3 viruses-14-01212-t003:** Relationship between clinicopathological features of OPSCCs and HPV presence.

Feature	HPV Presence			
	HPV-Negative	HPV-Positive	Total	*p*-Value
Gender				*p =* 0.24
Female	9 (50)	9 (50)	18	
Male	10 (32.25)	21 (67.74)	31	
Age				*p* = 0.3759
≤65 years	9 (32.14)	19 (67.85)	28	
>65 years	9 (45)	11 (55)	20	
Smoking Status				*p* = 0.95
Never	4 (33,33)	8 (66,77)	12	
Former	2 (40)	3 (60)	5	
Currently	5 (33,33)	10 (66,77)	15	
Keratinization				*p =* 0.11
Keratinized	16 (47.06)	18 (52.94)	34	
Non-keratinized	3 (20)	12 (80)	15	
Differentiation				*p =* 0.029 *
Poor	7 (29.17)	17 (70.83)	24	
Moderate	5 (31.25)	11 (68.75)	16	
Well	7 (77.77)	2 (22.22)	9	
Anatomical Site				*p =* 0.19
Palatine tonsils	7 (26.92)	19 (73.07)	26	
Tongue base	7 (50)	7 (50)	14	
Soft palate	5 (55.55)	4 (44.44)	9	
p16 IHC				*p =* 0.0004 ***
Positive	5 (17.85)	23 (82.14)	28	
Negative	14 (70)	6 (30)	20	

Gender, age, smoking status, keratinization, differentiation, anatomical site, and p16 IHC according to HPV presence. *: *p* ≤ 0.05: ***: *p* ≤ 0.001. *p* value < 0.5 was considered statistically significant.

## Data Availability

Not applicable.

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
