# Peer review of "Human Papillomavirus Detected in Oropharyngeal Cancers from Chilean Subjects"

_viruses, 2022, doi:10.3390/v14061212_

Round 1

Reviewer 1 Report

viruses-1720955-peer-review-v1

Comments and Suggestions for authors.

In this article, Carolina Oliva and colleagues evaluated the

detection of high-risk human papillomavirus in oropharyngeal tumors of Chilean subjects.

Although the research topic is interesting, this article has some issues that need to be solved.

Below are some suggestions, which would allow the article to be improved for publication.

1) It would be appropriate to include some epidemiological nods to HPV-related OPSCC in South America and possibly Chile as well.

2) Did the cases analyzed in this study have particular lifestyle habits (such as smoking or drinking) or had they previously had other HPV-related lesions?

3) In the abstract: "The HPV genotypes were identified through the sequencing of a generic L1 fragment" we suggest to rephrase the sentence and to describe appropriately in the methods the type of sequencing used and whether HPV L1 amplicons was generated by PCR amplification with degenerate consensus primers or not.

4) In materials and methods "HPV detection and genotyping" it would be better to include a reference for the use of general primers GP5 and GP6.

5) The legends in Figures and Tables 1 and 2 should be added as well as the statistical significance criteria.

6) In Table 1, "p16 IHQ" should be replaced by "p16 IHC".

7) Introduce in the text why samples were also sequenced for HPV33 and low-risk types such as HPV6.

8) In Table 3, it is suggested to recheck the presentation of data and add a column to compare data among HPV positives only; add legend. Given that there were 20 cases negative for expression of p16 surrogate marker of HPV, you should rephrase and expand the sentence "Finally, the three HPV6 cases were p16 negative, suggesting that this genotype is a bystander in clinical samples".

9) Figure 2 the array has imperfections it would be better to attach it in the supplemental data and include a representative image with adequate magnification of the 3 anatomical sites (Palatine Tonsils, Base of Tongue, Soft Palate). It would be best to include only representative images for each of the 3 anatomic sites analyzed, with appropriate magnification and highlighting a more magnified sub-area in each image. Add legend and describe the figure2.

10) Correct the numbering of the figures, there are two figures2: "Figure 2. Immunohistochemistry of p16 in OPSCC tissue arrays" and "Figure 2. E6 and E7 transcripts in OPSCCs".

11) In the figure "E6 and E7 transcripts in OPSCC" you should describe the analyzed cases shown and include the legend. It would be better to include all positive results including those for HPV33 and HPV6.

12) At the end of the discussion, "the 5 samples in which E6 and E7 transcripts were not detected, 4 of them were negative for P16, suggesting that even if the viral genome was present in the tissue, it was not playing an oncogenic role in the tumor." The subjects from whom the 4 P16-negative capions were derived, what characteristics did they have?

Author Response

Reviewer: In this article, Carolina Oliva and colleagues evaluated the detection of high-risk human papillomavirus in oropharyngeal tumors of Chilean subjects. Although the research topic is interesting, this article has some issues that need to be solved. Below are some suggestions, which would allow the article to be improved for publication.

1) It would be appropriate to include some epidemiological nods to HPV-related OPSCC in South America and possibly Chile as well.

Answer: We appreciate this observation. In the discussion section (second paragraph), we added more epidemiological data about HPV (+) OPSCCs in South America. Regarding Chile, there are no previously reported data, so we only have mentioned a previous study reporting HPV prevalence in the oral cavity.

Reviewer: 2) Did the cases analyzed in this study have particular lifestyle habits (such as smoking or drinking) or had they previously had other HPV-related lesions?

Answer: Taking this suggestion into account, we included smoking status in sections 2.1 and 3.1. In addition, Table 3 was expanded to include this data in relation to the presence of HPV. Unfortunately, no additional information was found in the clinical record.

Reviewer: 3) In the abstract: "The HPV genotypes were identified through the sequencing of a generic L1 fragment" we suggest to rephrase the sentence and to describe appropriately in the methods the type of sequencing used and whether HPV L1 amplicons was generated by PCR amplification with degenerate consensus primers or not.

Answer: In the abstract, the indicated phrase was deleted and replaced. In materials and methods, section 2.2, we added information about the amplification of the L1 fragment (including reference), and the sequencing method that is used by Macrogen.

Reviewer: 4) In materials and methods "HPV detection and genotyping" it would be better to include a reference for the use of general primers GP5 and GP6.

Answer: We included the requested reference (PMID: 9049358).

Reviewer: 5) The legends in Figures and Tables 1 and 2 should be added as well as the statistical significance criteria.

Answer: Suggestions included in the respective tables.

Reviewer: 6) In Table 1, "p16 IHQ" should be replaced by "p16 IHC".

Answer: This was corrected

Reviewer: 7) Introduce in the text why samples were also sequenced for HPV33 and low-risk types such as HPV6.

Answer: For HPV genotyping, the PCR products obtained after amplification of a 155 bp fragment of the L1 ORF are sequenced. In this way, it is possible to obtain the HPV genotype by comparing it with sequences previously reported and deposited in the GenBank platform. Therefore, we did not perform a specific sequencing for each HPV genotype. This point is clarified in methodology (Section 2.2).

Reviewer: 8) In Table 3, it is suggested to recheck the presentation of data and add a column to compare data among HPV positives only; add legend. Given that there were 20 cases negative for expression of p16 surrogate marker of HPV, you should rephrase and expand the sentence "Finally, the three HPV6 cases were p16 negative, suggesting that this genotype is a bystander in clinical samples".

Answer: Many thanks for these observations. We clarified in the text that most cases were HPV16 (described in Table 2), so comparisons among HPV positive cases for clinical features do not allow to determine statistical associations. On the other hand, the mentioned sentence was rephrased in the Results section as follow: “However, six p16 negative cases were HPV positive, including three HPV6 and three HPV16 cases. Conversely, five p16 positive cases were HPV negative”.

In the Discussion section, the interpretation of these findings was included, as follow: “Additionally, there were six p16 negative cases which were HPV positive, including 3 HPV6 cases, suggesting that this HPV genotype is a bystander. Additionally, other three cases were HPV16, suggesting the possibility that these cases are not HPV-driven cancers or that p16 was silenced by mutation or promoter hypermethylation in these patients, as previously reported in head and neck cancers [46,47]. Of note, five HPV negative cases, were p16 positive, suggesting the possibility of a false negative by PCR or that p16 was overexpressed by other mechanisms which are independent of E7 expression [48,49].”

Reviewer: 9) Figure 2 the array has imperfections it would be better to attach it in the supplemental data and include a representative image with adequate magnification of the 3 anatomical sites (Palatine Tonsils, Base of Tongue, Soft Palate). It would be best to include only representative images for each of the 3 anatomic sites analyzed, with appropriate magnification and highlighting a more magnified sub-area in each image. Add legend and describe the figure2.

Answer: Thank you for the observation. We deleted this figure and it was replaced by another that contains the three anatomical sites of the oropharynx, including hematoxylin eosin and p16 IHC staining, also detailing the histopathological characteristics in the figure legend.

Reviewer: 10) Correct the numbering of the figures, there are two figures2: "Figure 2. Immunohistochemistry of p16 in OPSCC tissue arrays" and "Figure 2. E6 and E7 transcripts in OPSCCs".

Answer: This was corrected

Reviewer: 11) In the figure "E6 and E7 transcripts in OPSCC" you should describe the analyzed cases shown and include the legend. It would be better to include all positive results including those for HPV33 and HPV6.

Answer: We appreciate your indication. We include a legend in figure 3 to provide more details. Regarding its other indications mentioned, we consider that it is not necessary to include the expression of HPV6 transcripts because this is a low-risk genotype that is not associated with the development of cancer in the oropharynx, on the other hand, only two cases were HPV33 positive, much lower than the 24 positive cases for HPV16.

Reviewer: 12) At the end of the discussion, "the 5 samples in which E6 and E7 transcripts were not detected, 4 of them were negative for P16, suggesting that even if the viral genome was present in the tissue, it was not playing an oncogenic role in the tumor." The subjects from whom the 4 P16-negative captions were derived, what characteristics did they have?

Answer: By evaluating distinctive features in these p16 cases, we wondered if these tumors could be tobacco associated. 2/4 were active smokers, however, the smoking history of the other two patients was not found in the clinical record. We did not find any additional distinguishing features of these 4 cases.

Reviewer 2 Report

he manuscript entitled “Human papillomavirus detected in oropharyngeal cancers from Chilean subjects” by Dr. Oliva reports on an evaluation of the presence of HPV in oropharyngeal cancers from Chilean subjects, in terms of viral DNA, E6 and E7 mRNA expressions as well as p16 expression. HPV DNA was detected in 61,2% of oropharyngeal carcinomas, the most prevalent genotype being HPV16 (80%). E6 and E7 transcripts were detected in 91.6% and 79.1% of the HPV16 positive specimens, respectively; p16 expression was positive in 58,3% of cases. Despite potentially interesting andconsidering that this is the first study investigating a cohort of OPSCC cases for HPV presence from Cile, the ms lack of novelty as various similar investigations and even meta analyses have been published in this topic PMID: 33115701. However, the manuscript is well written, while the experimental design is well performed. While tables are highly informative, figures quality/captions should be improved. The work will increase our knowledge on the role of HPV infection in the etiopathogenesis of OPSCC. Considering these aspects, I therefore recommend a major revision. I have several comments.

GENERAL COMMENTS
1.    HPV genome should be more well described. The genes encoded by the early region should be briefly mentioned as well as the long control region which contains the early promoter and regulatory element that play a role in viral replication and transcription. Main E6 and E7 targets should be briefly mentioned
2.    Several sentences are lacking in supporting references. Especially sections 2.2 and 2.4 of th emethods
3.    The total number of cases, alongside basic clinicopathological information, including age sex presence of co-morbidities, smoking vaccination status etc should be included in the fisrt section of the methods 
4.    . While tables are highly informative, figures quality/captions should be improved. The first and second figure of the ms are both quoted as figure 2. In addition sample distribution across the different stainings should be better described either in the caption or directly into the figure
5.    The authors should more deeply justify the possible overestimation of HPV prevalence in both tonsil tumor tissues and tongue tumors, found in the present study, and the relatively low rates in previous studies
6.    Limitations such as the relatively small sample size, lack of viral DNA load and E6 and E7 proteins determination should be included in the discussion

SPECIFIC
Abstract
1.    The fact that p16 is the surrogate marker for oncogenic HPV infection should be detailed 
2.    The number of investigated cases should be detailed 

Introduction
1.    “The virus infects the cells ….. epithelial tissue of the host [8].” HPV DNA sequences have also been identified in the respiratory cavity, such as the maxillary sinus. ( PMID:  33413530) this information should be included
2.    “More than 200 HPV …., or low risk of cancer.” This sentence is lacking in supporting references. PMID: 31134154
Results
1.    “The results showed that … for both oncoprotein transcripts.” English should be improved
2.    The fact that HPV33 is an oncogenic type should be detailed in the discussion DOI: 10.1016/j.biologicals.2019.02.001 

Discussion 
1.    Regarding references in the discussion 6, 17-20 the author should detail whether these studies evaluated the presence of HPV DNA only and/or additional HPV evidences, i.e., viral mRNA and proteins or p16

Author Response

Reviewer: The manuscript entitled “Human papillomavirus detected in oropharyngeal cancers from Chilean subjects” by Dr. Oliva reports on an evaluation of the presence of HPV in oropharyngeal cancers from Chilean subjects, in terms of viral DNA, E6 and E7 mRNA expressions as well as p16 expression. HPV DNA was detected in 61,2% of oropharyngeal carcinomas, the most prevalent genotype being HPV16 (80%). E6 and E7 transcripts were detected in 91.6% and 79.1% of the HPV16 positive specimens, respectively; p16 expression was positive in 58,3% of cases. Despite potentially interesting and considering that this is the first study investigating a cohort of OPSCC cases for HPV presence from Cile, the ms lack of novelty as various similar investigations and even meta-analyses have been published in this topic PMID: 33115701. However, the manuscript is well written, while the experimental design is well performed. While tables are highly informative, figures quality/captions should be improved. The work will increase our knowledge on the role of HPV infection in the etiopathogenesis of OPSCC. Considering these aspects, I therefore recommend a major revision. I have several comments.

Answer: We appreciate and consider your comments and corrections. Although a meta-analysis has been published on HPV positivity in oral and oropharyngeal carcinoma (PMID 35432780), only Brazil reports oropharyngeal cases. An article from Colombia is titled in English as oropharynx, however, this is not correct since the original title in Spanish refers to oral cavity and furthermore, when going into detail of the methodology, only cases of oral cavity were considered (http://www.scielo.org.co/scielo.php?script=sci_arttext&pid=S0120-41572016000600003). Therefore, considering that there is only one country in the region with details of the oropharynx, we believe that this manuscript is a contribution to this matter.

Reviewer:

GENERAL COMMENTS

  1. HPV genome should be more well described. The genes encoded by the early region should be briefly mentioned as well as the long control region which contains the early promoter and regulatory element that play a role in viral replication and transcription. Main E6 and E7 targets should be briefly mentioned

Answer: We added the information indicated in the second paragraph of the introduction, except for the main targets of E6 and E7 that had been mentioned in the penultimate paragraph of the introduction.

Reviewer: 2.    Several sentences are lacking in supporting references. Especially sections 2.2 and 2.4 of the methods

Answer: We include the following references: PMID: 7653107, 11351039, 9049358, 31436299

Reviewer: 3.    The total number of cases, alongside basic clinicopathological information, including age sex presence of co-morbidities, smoking vaccination status etc. should be included in the first section of the methods

Answer: We included the information requested in section 2.1, however, we did not find information about comorbidity. Regarding vaccination, in Chile, the national vaccination against HPV was introduced in 2014 for girls between 9 and 10 years old, and since 2019 for boys between 9 and 10 years old, therefore, these patients do not present immunization against the virus.

Reviewer: 4. While tables are highly informative, figures quality/captions should be improved. The first and second figure of the ms are both quoted as figure 2. In addition sample distribution across the different stainings should be better described either in the caption or directly into the figure

Answer: We deleted this figure and it was replaced by another that contains the three anatomical sites of the oropharynx, including hematoxylin eosin and p16 IHC staining, also detailing the histopathological characteristics in the figure legend.

Reviewer: 5.    The authors should more deeply justify the possible overestimation of HPV prevalence in both tonsil tumor tissues and tongue tumors, found in the present study, and the relatively low rates in previous studies.

Answer: In the discussion, we expand on the prevalence in South America and the possible factors that explain the differences reported.

Reviewer: 6.    Limitations such as the relatively small sample size, lack of viral DNA load and E6 and E7 proteins determination should be included in the discussion

Answer: We include in the discussion some limitations, such as the small size of soft palate biopsies, and the possibility of degradation mainly of RNA in FFPE. Despite these limitations, to ensure the integrity of the genetic material in the assays, β-Globin was evaluated for DNA (we excluded 2 negative samples for this gene) and also β-Actin for RNA. Additionally, primers that target short fragments of oncoproteins were used to optimize detection.

Reviewer: SPECIFIC

Abstract

  1. The fact that p16 is the surrogate marker for oncogenic HPV infection should be detailed

Answer: Added the data in the abstract and reaffirmed in the introduction.

Reviewer: 2.    The number of investigated cases should be detailed

Answer: Added in abstract.

Reviewer: Introduction

  1. “The virus infects the cells ….. epithelial tissue of the host [8].” HPV DNA sequences have also been identified in the respiratory cavity, such as the maxillary sinus. ( PMID:33413530) this information should be included

Answer: We include in the introduction information regarding the presence of HPV in other carcinomas of the head and neck, including paranasal cavities.

Reviewer: 2.    “More than 200 HPV …., or low risk of cancer.” This sentence is lacking in supporting references. PMID: 31134154

Answer: Corrected.

Reviewer: Results

  1. “The results showed that … for both oncoprotein transcripts.” English should be improved

Answer: The sentence indicated was reformulated.

Reviewer: 2.    The fact that HPV33 is an oncogenic type should be detailed in the discussion DOI: 10.1016/j.biologicals.2019.02.001

Answer: This was included in the discussion

Reviewer: Discussion

  1. Regarding references in the discussion 6, 17-20 the author should detail whether these studies evaluated the presence of HPV DNA only and/or additional HPV evidences, i.e., viral mRNA and proteins or p16

Answer: Thanks for this suggestion. Due to the high variability of methods, we prefer to include a paragraph in the discussion addressing the issue of different detection methods and the heterogeneity of the criteria used to determine if a tumor is HPV positive.

Round 2

Reviewer 2 Report

The ms can be accepted in the present form